# Efficient Helium Separation with Two-Dimensional Metal–Organic Framework Fe/Ni-PTC: A Theoretical Study

**DOI:** 10.3390/membranes11120927

**Published:** 2021-11-26

**Authors:** Jingyuan Wang, Yixiang Li, Yanmei Yang, Yongqiang Li, Mingwen Zhao, Weifeng Li, Jing Guan, Yuanyuan Qu

**Affiliations:** 1School of Physics, Shandong University, Jinan 250100, China; wjy96@mail.sdu.edu.cn (J.W.); lyx1113@mail.sdu.edu.cn (Y.L.); yqli@sdu.edu.cn (Y.L.); zmw@sdu.edu.cn (M.Z.); lwf@sdu.edu.cn (W.L.); 2College of Chemistry, Chemical Engineering and Materials Science, Shandong Normal University, Jinan 250014, China; yym@sdnu.edu.cn

**Keywords:** gas separation, metal–organic framework, membrane-based separation, molecular dynamics simulation, first-principles calculations

## Abstract

Helium (He) is one of the indispensable and rare strategic materials for national defense and high-tech industries. However, daunting challenges have to be overcome for the supply shortage of He resources. Benefitted from the wide pore size distribution, sufficient intrinsic porosity, and high specific surface area, metal–organic framework (MOF) materials are prospective candidates for He purification in the membrane-based separation technology. In this work, through first-principles calculations and molecular dynamics (MD) simulations, we studied the permeability and filtration performance of He by the newly synthesized two-dimensional Fe-PTC MOF and its analogue Ni-PTC MOF. We found that both Fe-PTC and Ni-PTC have superior high performance for He separation. The selectivity of He over N_2_ was calculated to be ~10^17^ for Fe-PTC and ~10^15^ for Ni-PTC, respectively, both higher than most of the previously proposed 2D porous membranes. Meanwhile, high He permeance (10^−4^~10^−3^ mol s^−1^ m^−2^ Pa^−1^) can be obtained for the Fe/Ni-PTC MOF for temperatures ranging from 200 to 500 K. Therefore, the present study offers a highly prospective membrane for He separation, which has great potential in industrial application.

## 1. Introduction

As a rare strategic material, helium (He) is urgently needed in the medical, scientific research, and aerospace industries [1,2,3]. Albeit being the second most abundant element in the universe, He is still scarce in the atmosphere. Currently, the main source of He is the by-product in the exploitation of natural gas [4,5]. Purification of He with membrane-based separation technology has become a promising method, because of its low energy consumption, eco-friendly nature, simple operation, and other advantages [6,7,8,9,10].

The recent quick advances in the development of nanoporous two-dimensional (2D) structures have provided superior membrane candidates for He separation, in the merit of their intrinsic porosity, atomic thickness, and high specific surface area. For instance, graphene [9], silicene [11], and MoS_2_ [12] with drilling holes have been proposed for He separation and demonstrated to be successful. However, the requirement of precise treatment to these materials to introduce properly-sized hole structure becomes a technique hindrance, thus 2D structures with intrinsically distributed nanopores have attracted more attention which are highly expected to be efficient filtration membranes. For example, in 2015, the experimentally available g-C_3_N_4_ membrane has been demonstrated to possess high selectivity (10^7^~10^65^ at 300 K) for separating He from impure gas molecules (H_2_, N_2_, CO, CH_4_, Ne, and Ar) in natural gas [13]. Then, Wang et al. theoretically predicted high He selectivity (10^2^~10^38^ at 300 K) and permeance (1.0 × 10^3^ GPU) of the monolayer CTF-0 membrane for He treatment [14]. Very recently, Liu et al. has shown that the nano-porous g-C_2_O membrane is a promising membrane for He separation with high selectivity (30~10^28^ at 300 K) and permeance (1.03 × 10^7^ GPU) [15]. In addition, other intrinsically porous 2D materials, including C_2_N [16], IGP [17] and graphenylenee-1 [18], have also shown good performance for He separation. It should be noted that the trade-off between the selectivity and permeance is inherent in the system with porous structure, as large pores are favored by high permeance and small pores are advantageous to the selectivity of small molecules like He. Therefore, searching for a membrane with properly-sized pores for He separation is of great importance to achieve the goal of optimized balance of high permeance with sufficiently high selectivity.

Metal–organic frameworks (MOFs) are a new class of nanoporous materials, which is constituted of metal nodes connected by organic linkers [19,20]. Due to the high diversity of functional groups and ligands, the structure of MOFs can be tailored during the synthesis process, resulting in a large variety of structures with diverse geometries, pore sizes, and functional groups [21]. To date, a great deal of MOFs has been reported with excellent properties, such as high pore volume, large surface area, wide range of pore size from micro to meso scale, and reasonable thermal and mechanical stability. Based on these properties, application studies of MOFs for gas storage and separation [22,23,24,25,26], drug delivery [27], and catalysis [25,28] are widely explored. In particular, because of the rich porous structure, many MOFs have been proposed to be effective membrane materials in gas separation. Ranjan et al. [29] showed that microporous metal–organic framework (MMOF) membranes exhibit industrially acceptable selectivity of 23 for H_2_/N_2_ separation at 363 K. Cao et al. [30] demonstrated that the Cu−BTC membrane can be used for He separation with moderate selectivities and permeance for He over CO_2_, N_2_, and CH_4_, comparable to previously reported MOF membranes. In 2018, a highly crystalline ZIF-68 membrane was synthesized by Kasik et al., which offered a high He permeance of 2.6 × 10^−7^ mol s^−1^ m^−2^ Pa^−1^ [31]. The above-mentioned reports are all based on three-dimensional (3D) MOF structures, and relative application studies of 2D MOFs in gas separation are rare.

In 2018, Dong et al. successfully synthesized the black polycrystalline 2D Fe-PTC MOF by solvothermal method, which exhibits a high electrical conductivity of 10 S cm^−1^ (300 K) and a ferromagnetism (<20 K), implying its potential as a ferromagnetic semiconductor for application in spintronics [32]. It is noticed that the pore size of the Fe-PTC MOF is about 5.65 Å [33], similar to that of IGP (5.47 Å) [17] which has good performance on He separation. In addition, by replacing the transition metal atoms in the Fe-PTC MOF, a group of analogues can be obtained with similar pore sizes [33]. Given their atomic thickness and proper pore sizes, the He purification of this group of 2D MOF is highly expected.

In this work, using combined first-principles calculations and MD simulations, we theoretically demonstrated that the 2D Fe-PTC MOF and its analogues Ni-PTC MOF can efficiently separate He from natural gas and inert gases. The He permeance of Fe/Ni-PTC MOF reached ~6 × 10^−4^ mol s^−1^ m^−2^ Pa^−1^. It is worth mentioning that this permeability is five orders of magnitude higher than the industrially acceptable value (6.9 × 10^−9^ mol s^−1^ m^−2^ Pa^−1^), indicating its potential in industrial application. Moreover, the He/N_2_ selectivity is estimated to be 10^15^~10^17^, higher than most of the previously proposed 2D porous membranes. Hence, our results provide a new type of competitive He separation membranes with promising industrial applications.

## 2. Computational Methods

The Vienna ab initio simulation program (VASP) in the framework of density functional theory (DFT) is used to perform all first-principles calculations [34,35]. The projected-augmented wave (PAW) potential [36] was employed to describe the electron–ion interaction. The Perdew–Burke–Ernzerhof (PBE) functional under the generalized gradient approximation (GGA) was used to describe the electron–electron interaction [37]. The weak van der Waals interaction between the gas molecules and the Fe/Ni-PTC membrane was corrected by the Grimme method (DFT−D2) [38]. The energy cutoff was set to be 500 eV and iterative convergences of energy and force were 10^−5^ eV and 10^−2^ eV/Å. In order to prevent the interaction between adjacent images, a 20 Å vacuum zone was applied in the z direction. A k-point mesh of 5 × 5 × 1 was chosen for sampling the Brillouin zone for the unit cell [39]. The convergence of the system energy with respect to energy cutoff and the k-point mesh has been tested and shown in Appendix A. For the pristine membrane, fully structural optimization was performed. For the adsorption interaction between gases and membranes, the cell vectors were fixed as negligible influence (membrane areas or adsorption energies) was observed upon gas adsorption (Appendix A). Meanwhile, the z-coordinates of the membranes were frozen to maintain a planar surface, while the x- and y-coordinates of the membrane and the coordinates of the gas molecules were fully relaxed. The cohesive energies of these two MOF membranes and the adsorption energies of gas molecules adsorbed on a membrane are all obtained from single point energy calculation. The climbing image nudged elastic band (cNEB) method was adopted for searching the minimum energy pathway of the gas molecule passing through the 2D MOF membrane [40]. In the transition state calculation, we used convergence criteria of energy and force of 10^−5^ eV and 0.02 eV/Å. Moreover, 5 images were used along the minimum energy pathway, and the z-coordinates of the membrane were kept frozen to maintain a planar surface.

The molecular dynamics (MD) simulations were performed within the GROMACS package using the universal force field (UFF) [41,42]. The three partial charge models for CO_2_ and N_2_ molecules were used to account for the quadrupole distinction of the CO_2_ and N_2_ molecules (Appendix A) [43,44]. The atomic charges of CH_4_ and CO were adopted from previous work [17], and are shown in Appendix A. The atomic charges of Fe/Ni-PTC membranes were obtained from the ESP fitting based on Merz–Kollman scheme by Gaussian 09 at the B3LYP/6-31g(d) level [45]. The detailed charge information for the Fe/Ni-PTC is shown in Appendix A in the ESI. In the molecular dynamics simulation system, seven prototype gas molecules, including He, Ne, CO_2_, N_2_, Ar, CO, and CH_4_ (50 for each molecule), were randomly placed into a simulation box sandwiched by a graphene sheet and a Fe/Ni-PTC membrane. The graphene sheet acts as a piston in the system, which was applied with a constant force of 115.1 kJ mol^−1^ nm^−1^ (equivalent to a constant pressure of 100 Bar), while the coordinates of the Fe/Ni-PTC membrane were kept frozen. The volume of the simulation box is 4.068 nm × 4.698 nm × 90 nm, and 3D periodic boundary conditions were applied.

The potential of mean force (PMF) [46] for He and N_2_ gases are calculated by means of the umbrella sampling method. Sixteen sampling windows with 0.1 nm intervals were taken on one side of the MOF membrane, and the total sampling length was 1.5 nm. For each sampling point, the position of the selected gas molecule was restrained using a harmonic potential along the reaction coordinate (z direction). The system was firstly equilibrated for 1 ns, and then the PMF was generated from 20 ns simulated force data in the canonical ensemble using the weighted histogram analysis method [47].

## 3. Results and Discussion

### 3.1. The Structural Stability of Fe/Ni-PTC

The metal−PTC 2D MOF has several members depending on the specific types of transition metal in the nodes. In this study, we took Fe-PTC and Ni-PTC as representative models to study their filtration performance for He separation. The optimized structures at DFT level are shown in Figure 1, where the lattice constants for Fe-PTC and Ni-PTC are determined to be *a* = *b* = 1.356 nm with the angle between them α = 120°, in good agreement with previous studies [33]. The diameters of the pore are 5.65 Å and 5.67 Å for Fe-PTC and Ni-PTC, respectively. For the Fe-PTC, the lengths of C-C bond, C-S bond, and Fe-S bond are 1.450 Å, 1.729 Å, and 2.094 Å, while for the Ni-PTC, that of the C-C, C-S, and Ni-S bonds are 1.449 Å, 1.710 Å, and 2.128 Å, which are all consistent with previously reported values [32,33]. The cohesive energy is used to examine the structural stability of the Fe/Ni-PTC monolayer, which represents the energy required to break the monolayer down into individual atoms, as defined by [48]:(1)$$E_{coh}=\left(n_{1}E_{c}+n_{2}E_{s}+n_{3}E_{Fe/Ni}-E_{mem}\right)/\left(n_{1}+n_{2}+n_{3}\right)$$
where *E_c_*_,_ *E_s_*, *E_Fe/Ni_*, and *E_mem_* denote the energies of a single C atom, a single S atom, a single Fe/Ni atom, and the total energy of the Fe/Ni-PTC monolayer, respectively; *n*_1_, *n*_2_, and *n*_3_ denote the total numbers of the C atoms, the S atoms, and the Fe/Ni atoms in the unit cell. The cohesive energy of the monolayer Fe-PTC was computed to be 7.10 eV/atom while that of Ni-PTC was 7.21 eV/atom, slightly higher than that of Fe-PTC, indicating its high energetic stability and the possibility of experimental synthesis. Moreover, a previous study has shown that both Fe-PTC and Ni-PTC are all thermally stable [33], which is crucial for gas separation.

### 3.2. DFT Energy Barriers

In this study, several prototype gas molecules (He, Ne, CO_2_, CO, N_2_, Ar, and CH_4_) from the by-product during natural gas exploitation [49] were selected to assess the He filtration performance of Fe-PTC and Ni-PTC membranes. Prior to exploring the energy barrier of an individual gas molecule penetrating the Fe/Ni-PTC membrane, the adsorption ability of Fe/Ni-PTC monolayers to each gas molecule has been investigated. The adsorption energy between the Fe/Ni-PTC membrane and a gas molecule can be expressed by the following formula:(2)$$E_{ad}=E_{gas+mem}-E_{gas}-E_{mem}$$
where *E_gas_* represents the energy of an individual gas molecule, *E_mem_* represents the energy of a pristine membrane, both of which were obtained from the single point energy calculations computed from the geometry-optimized structures of the isolated components, and *E_gas+mem_* represents the total energies of a gas molecule adsorbed on the membrane at the most stable adsorption state. The adsorption energy and adsorption height of the most stable state for each molecule on the Fe-PTC and Ni-PTC membranes are summarized in Table 1 and Table 2. It can be seen that the adsorption energy ranges from −2 to −17 kJ mol^−1^ and the adsorption height is in the range from 2.8 to 3.0 Å, indicating that the gas molecules are physically adsorbed on the Fe/Ni-PTC membrane. Among these molecules, the He molecule has the weakest interaction with the Fe/Ni-PTC (−2.60/−2.31 kJ mol^−1^), which may facilitate the diffusion of He through the Fe/Ni-PTC membrane. The illustration of the adsorption state of each gas molecule on the Fe/Ni-PTC membrane can be found in Appendix A, where guest molecules uniformly prefer to adsorb on the pore area. Thus, the transition metal cations (Fe or Ni) do not play an important role on stabilization of the guest molecules, because no direct interactions between them were formed.

With the most stable adsorption state, the translocating pathway of an individual gas molecule through the Fe/Ni-PTC membrane was searched by the cNEB approach [40]. The energy barrier (*E_b_*) of a gas molecule passing through the membrane can be defined as [50]:(3)$$E_{b}=E_{TS}-E_{SS}$$
where *E_TS_* denotes the energy of a gas molecule and the Fe/Ni-PTC membrane at the transition state (namely, the energy saddle point) and *E_SS_* represents that at the most stable adsorption state. The energy profiles of all seven gas molecules passing through the Fe/Ni-PTC membrane are plotted in Figure 1c,d respectively, and the energy barriers are summarized in Table 1 and Table 2. It is obvious that the energy barriers for seven gases are increased following the order of He < Ne < CO_2_ < N_2_ < Ar < CO < CH_4_, where He has the lowest energy barriers to pass through the Fe-PTC (14.64 kJ mol^−1^) and Ni-PTC (13.00 kJ mol^−1^) membranes, respectively. The electron density difference maps for the transition state for both gas-Fe-PTC and gas-Ni-PTC systems in Appendix A also show that the coupling between He and the membranes are the weakest, while that between CH_4_ and the membranes are the strongest, because the most significant electron transfer happened between CH_4_ and the membranes. Moreover, for electron density difference maps at the states near the transition states in Appendix A, dipole-shaped interactions can be clearly observed for CO_2_ and N_2_, which are induced by the inherent electric field of the membranes. These results reveal that He is more inclined to be separated from other gas molecules by the Fe/Ni-PTC membrane. 

### 3.3. Electron Density Isosurfaces

To better understand the different barriers of Fe/Ni-PTC to seven gas molecules, we then calculated the electron density of gas molecules interacting with the Fe-PTC monolayer, because electron density overlap between them results in repulsive interaction between the gas molecules and the membrane [13]. As shown in Figure 2, with isovalue of 0.01 e Å^−3^, there is no obvious electron density overlap between the He molecule and the Fe-PTC membrane, which is well consistent with the lowest energy barrier for He diffusing through the Fe-PTC membrane. For other molecules, clear electron density overlap is observed, resulting in higher energy barriers. The electron overlap between CH_4_ and Fe-PTC membrane is the most significant, thus the highest energy barrier was obtained for CH_4_ translocating the Fe-PTC membrane. The electron density isosurfaces of the gas molecule interacting with the Ni-PTC monolayer are depicted in Appendix A, where a similar trend has been observed.

### 3.4. DFT Selectivity of He

Based on the energy barriers of gas molecule translocating the Fe/Ni-PTC membrane, the selectivity of the He molecule to other gases can be evaluated via the Arrhenius equation [51]:(4)$$S_{He/gas}=\frac{r_{He}}{r_{gas}}=\frac{A_{He}e^{-E_{He}/RT}}{A_{gas}e^{-E_{gas}/RT}}$$
where *r*, *A*, and *E* represent the diffusion rate, the diffusion prefactor, and the diffusion energy barrier summarized in Table 1. The diffusion prefactor of all gases in this work was assumed to be 10^11^ s^−1^, following previous studies [52,53]. The diffusion rates of all the gas molecules and the selectivity of He over other gas molecules as a function of temperature are shown in Figure 3. With the increase of temperature, the diffusion rate of gas molecules uniformly rises, which can be attributed to the increased kinetic energy of gas molecules in the high-temperature region. In addition, the selectivity of He over other molecules of the Fe/Ni-PTC membrane decreases with the increase of temperature.

### 3.5. MD Simulations on He Filtration

For industrial He separation, the operation is normally carried out at room temperature [54]. Therefore, MD simulations were conducted to verify the separation performance of Fe/Ni-PTC membranes at 300 K. In the simulation model shown in Figure 4a, 50 He, 50 Ne, 50 Ar, 50 CO, 50 CO_2_, 50 N_2_, and 50 CH_4_ were mixed and randomly distributed between a graphene piston and a Fe/Ni-PTC membrane. Three parallel trajectories of 90 ns have been generated and the results are plotted in Figure 4b,c. It can be seen that during the 90 ns simulation, consecutive events of He passing through the Fe/Ni-PTC membrane were monitored and after 90 ns, nearly all of the He molecules had penetrated through the Fe/Ni-PTC membrane. In contrast, other molecules remained between the two membranes as indicated by the green lines. The insets show the He flow rates perpendicular to the membranes, which decrease with time as increasingly more He molecules have escaped away. Therefore, the results from the MD simulation demonstrated that Fe/Ni-PTC is highly selective to He over other gas molecules. The representative trajectory snapshots are shown in Appendix A in the ESI.

In addition to selectivity, He permeance is another indicative factor to judge the separation efficiency of a membrane. Thus, we further calculated the He permeance of Fe/Ni-PTC membrane at temperatures ranging from 200–500 K, following the method reported in a previous work [55] which can be found in the ESI. The results of He permeance are listed in Table 3. It can be seen that as the temperature increases, the permeance gradually increased from 10^−4^ to 10^−3^ mol s^−1^ m^−2^ Pa^−1^, which is attributed to the increased kinetic energy of He particles at high temperatures. At room temperature of 300 K, the permeance of He reached ~ 6 × 10^−4^ mol s^−1^ m^−2^ Pa^−^^1^ for both Fe-PTC and Ni-PTC membrane, which is five orders of magnitude higher than the industrially acceptable value (6.79 × 10^−9^ mol s^−1^ m^−2^ Pa^−1^) [56], implying the plausibility of the Fe/Ni-PTC membrane for the industrial applications.

### 3.6. Comparison between Membranes

After the pre-treatment of natural gas, the major component in the remaining gas stream is N_2_. Thus, the selectivity of the Fe/Ni-PTC for He over N_2_ is of great significance in practical applications. We compared the He permeance and the He/N_2_ selectivity of both Fe-PTC and Ni-PTC membrane with previously proposed nanoporous membranes to evaluate the He separation efficiency of the two MOFs. As shown in Table 4, it is obvious that although the Fe/Ni-PTC has relatively lower He permeance, the He/N_2_ selectivity of Fe/Ni-PTC MOF is significantly higher than those of g-C_2_O, C_2_N and 6N-PG membranes. While compared to the CTF-0 membrane, the selectivity of Fe/Ni-PTC MOF is lower, but the permeance is three orders of magnitude higher. It should be noted that compared to the IGP membrane, with almost the same permeance, the He selectivity of Fe/Ni-PTC is higher by five orders of magnitude, indicating its superior performance for He harvest.

### 3.7. Free Energy Barrier Analysis

The above results have demonstrated that the Fe/Ni-PTC membrane have excellent selectivity and permeance for He separation, both in the high selectivity and permeance. To get a deeper insight into the improved He/N_2_ selectivity and He permeance at room temperature, we have estimated the free energy barriers by calculating the PMF of He and N_2_ passing through the Fe/Ni-PTC membrane and the results are summarized in Figure 5. As can be seen, the free energy barrier values for He penetrating Fe/Ni-PTC membrane are 5.62 and 5.18 k_B_T, respectively, which are much lower than those of N_2_ (73.8 and 66.2 k_B_T), resulting in both the high He/N_2_ selectivity and high He permeance. It is worth noting that compared with the energy barriers based on DFT calculation for N_2_ (117.61/100.95 kJ mol^−1^, equivalent to 47.2/40.5 k_B_T for Fe-PTC and Ni-PTC respectively), the free energy barriers obtained from PMF analysis are increased. This is because the temperature-related conformational entropy for rod-like molecules such as N_2_ is explicitly considered. 

## 4. Conclusions

To conclude, we performed combined first-principles calculations and MD simulations to theoretically investigate the feasibility of the 2D Fe/Ni-PTC MOF materials for He separation from natural gas and other noble gases. High permeance (~6 × 10^−4^ mol s^−1^ m^−2^ Pa^−1^) and He/N_2_ selectivity (10^15^~10^17^) of Fe/Ni-PTC MOF are predicted, superior to most of the previously proposed 2D porous membranes. Moreover, quantitative free energy analysis indicates that the free energy barriers for He passing through the Fe/Ni-PTC MOF are only ~5 k_B_T, which is significantly lower than that of N_2_ (~70 k_B_T), resulting in the high selectivity for He separation at room temperature. Overall, our results have demonstrated that the newly synthesized 2D Fe-PTC and its analogue Ni-PTC are promising membranes for He separation, which might have great prospects in industrial production.

## Figures and Tables

**Figure 1 membranes-11-00927-f001:**
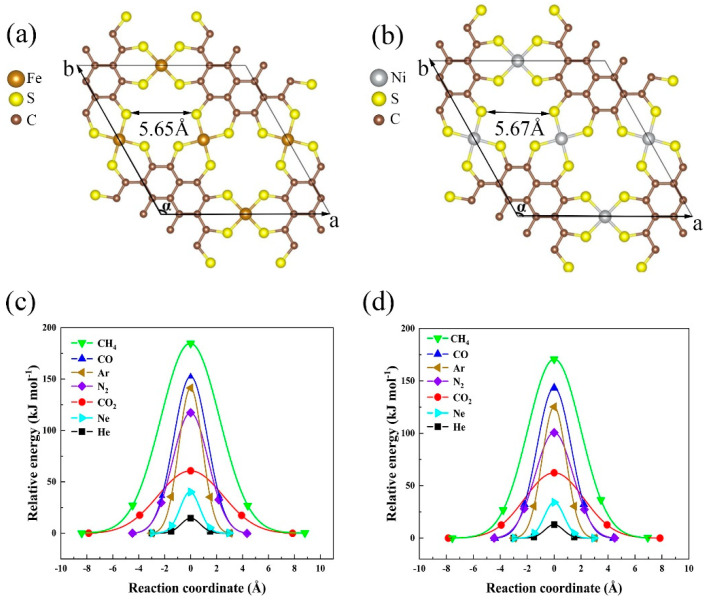
The top views of the optimized unit cell of the (**a**) Fe-PTC and (**b**) Ni-PTC monolayer, where $\vec{a}$ and $\vec{b}$ represent the cell vectors and α represents the angle between them. The energy profiles for seven prototype gas molecules translocating the (**c**) Fe-PTC and (**d**) Ni-PTC membranes.

**Figure 2 membranes-11-00927-f002:**
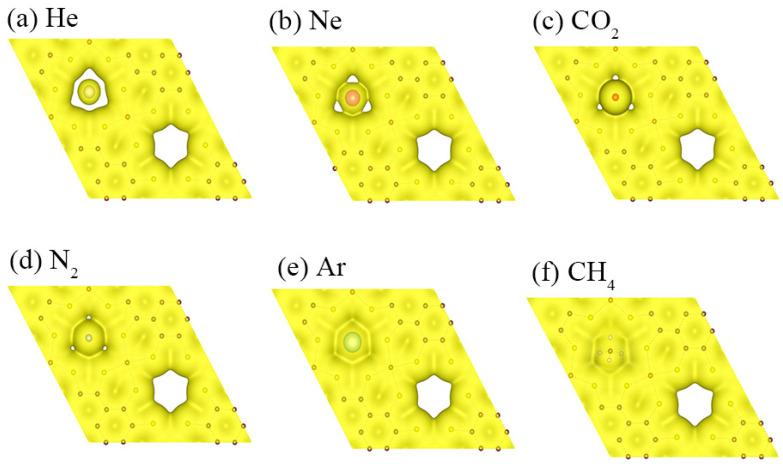
Electron density isosurfaces for the gas molecules: (**a**) He, (**b**) Ne, (**c**) CO_2_, (**d**) N_2_, (**e**) Ar and (**f**) CH_4_ translocating the Fe-PTC membrane (isovalue of 0.01 Å^−3^).

**Figure 3 membranes-11-00927-f003:**
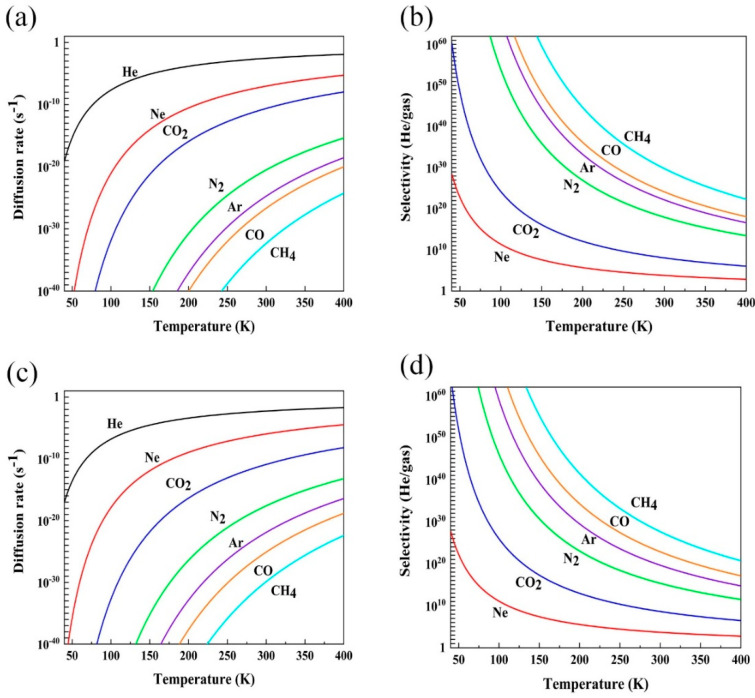
(**a**) The diffusion rates and (**b**) the selectivity for gases to pass through the Fe-PTC membrane as a function of temperature; (**c**) The diffusion rates and (**d**) the selectivity for gases to pass through the Ni-PTC membrane as a function of temperature. Different colors denote different gas molecules.

**Figure 4 membranes-11-00927-f004:**
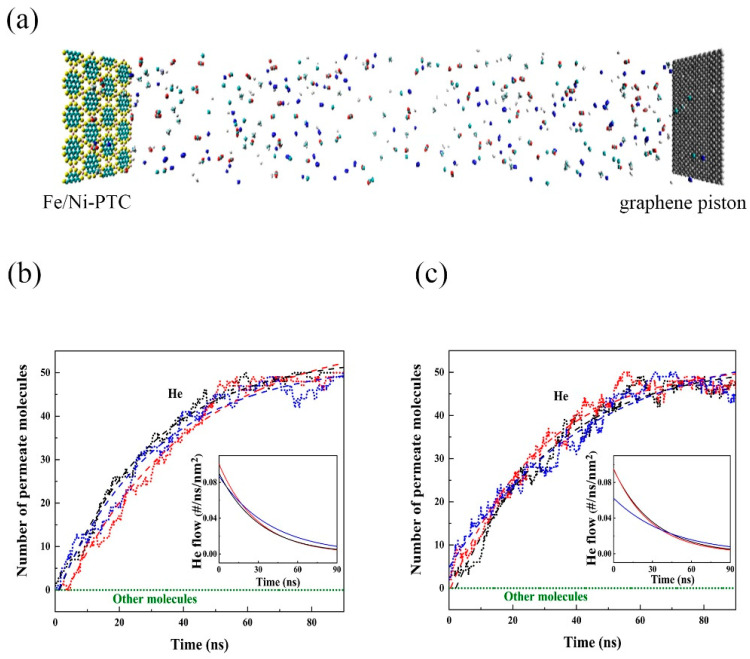
(**a**) The snapshot of the simulation model; pure He permeation by (**b**) Fe-PTC and (**c**) Ni-PTC membranes at 300 K, where individual results obtained from three independent trajectories are represented by red, blue, and black dots. The dashed lines are the numerical fitted results. The insets show the gas flow rates calculated from the numerical fitted results.

**Figure 5 membranes-11-00927-f005:**
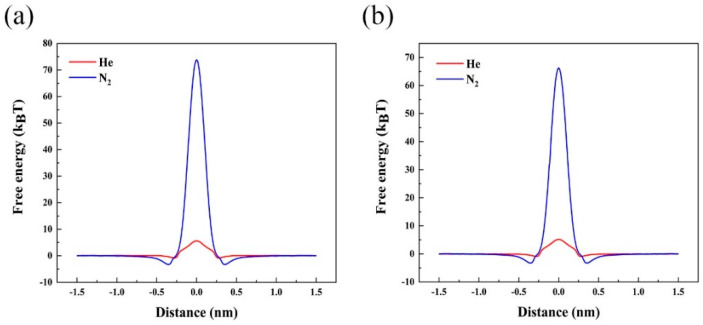
The PMF of He and N_2_ passes through the (**a**) Fe-PTC and (**b**) Ni-PTC membranes by umbrella sampling.

**Table 1 membranes-11-00927-t001:** The adsorption height (H_0_), the corresponding adsorption energy (E*_ad_*) between the gas molecule and the Fe-PTC monolayer, and the energy barriers (E*_b_*) of the seven gas molecules translocating the Fe-PTC monolayer.

Gas Molecule	H_0_ (Å)	E*_ad_* (kJ mol^−1^)	E*_b_* (kJ mol^−1^)
He	3.000	−2.60	14.64
Ne	2.901	−4.43	40.26
CO_2_	3.337	−9.25	60.87
N_2_	2.836	−13.20	117.61
Ar	2.987	−10.98	141.78
CO	3.000	−13.97	152.96
CH_4_	2.950	−14.74	185.42

**Table 2 membranes-11-00927-t002:** The adsorption height (H_0_), the corresponding adsorption energy (E*_ad_*) between the gas molecule and the Ni-PTC monolayer, and the energy barriers (E*_b_*) of the seven gas molecules translocating the Ni-PTC monolayer.

Gas Molecule	H_0_ (Å)	E*_ad_* (kJ mol^−1^)	E*_b_* (kJ mol^−1^)
He	2.997	−2.31	13.00
Ne	3.000	−4.43	34.19
CO_2_	2.929	−6.84	62.51
N_2_	2.918	−13.00	100.95
Ar	3.000	−10.79	125.61
CO	2.999	−13.87	143.91
CH_4_	2.683	−16.28	171.36

**Table 3 membranes-11-00927-t003:** He permeance (mol s^−1^ m^−2^ Pa^−1^) at the temperature range of 200–500 K.

Membrane	Temperature (K)	Permeance
Fe-PTC	200	2.9 × 10^−4^
	300	6.3 × 10^−4^
	400	8.9 × 10^−4^
	500	1.3 × 10^−3^
Ni-PTC	200	3.6 × 10^−4^
	300	6.6 × 10^−4^
	400	9.2 × 10^−4^
	500	1.5 × 10^−3^

**Table 4 membranes-11-00927-t004:** Comparison results of He permeance (GPU) and the He/N_2_ selectivity between Fe/Ni-PTC MOF and previously proposed porous membranes at 300 K. (1 GPU = 3.3 × 10^−10^ mol s^−1^ m^−2^ Pa^−1^).

Membrane	Permeance	Selectivity
Fe-PTC ^1^	1.9 × 10^6^	9.1 × 10^17^
Ni-PTC ^1^	2.1 × 10^6^	2.2 × 10^15^
g-C_2_O ^2^	1.0 × 10^7^	1.5 × 10^6^
IGP ^3^	2.0 × 10^6^	1.0 × 10^12^
CTF-0 ^4^	1.0 × 10^3^	2.0 × 10^27^
C_2_N ^5^	1.0 × 10^7^	3.0 × 10^12^
6N-PG ^6^	6.9 × 10^7^	6.0 × 10^8^

^1^ This study. ^2^ Ref. [15]. ^3^ Ref. [17]. ^4^ Ref. [14]. **^5^** Ref. [16]. **^6^** Ref. [57].

## Data Availability

The data that support the findings of this study are available from the corresponding author upon reasonable request.

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
