# Peer review of "Efficient Helium Separation with Two-Dimensional Metal–Organic Framework Fe/Ni-PTC: A Theoretical Study"

_membranes, 2021, doi:10.3390/membranes11120927_

Round 1

Reviewer 1 Report

In this manuscript authors theoretically studied the  permeability and filtration performance of He by the newly synthesized two-dimensional Fe-PTC MOF and its analogue Ni- 20, through first-principles 
calculations and molecular dynamics (MD) simulations. The manuscript is well written and can be accepted after MINOR revision.

  1. Authors should re-write the title mentioning that the theoretical investigation was conducted.
  2. Authors should cite the work where they/or other authors synthesized the 2D Fe-PTC MOF and its analogues.
  3. It is better to relate the surface area of prepared MOF with gas permeation intensity, if possible.
  4. Authors make clear sentences about energy calculations either calculated from geometry optimization or from single point energy calculation, where made.

Reviewer 2 Report

The paper details the computational simulations of the permeability and selectivity towards He separation of two membranes. The authors employed DFT and forcefield-based MD simulations to collect the necessary data for their conclusions. Particularly, the MD simulations provide interesting results about the permeability/selectivity of both membranes. This article may be useful for future experimental and computational studies in the research area of membrane separation. I recommend publication after the clarification of the following points:

Computational Methods

1) Did the authors compute the spin polarization of the Fe and Ni atoms for the DFT calculations?

2) How did the authors test the energy cutoff of the planewaves? Did they converge the cutoff with the total energy, forces, stress or else?

3) Did the authors optimize the vectors along the membranes?

4) It is recommended that the authors inform the dimensions of the membranes in the Computational Methods section. The length of the vectors and the angle between them is enough for the reader to reproduce the calculations.

5) I fail to understand why some atoms were frozen during the cNEB simulation. It would be better to illustrate this geometry constraint with a new figure.

6) The authors should specify the number of images used along the minimum energy path and the convergence criteria for cNEB.

Results and Discussion

7) I see that the authors used the comparison between the cohesion energies of both membranes to demonstrate their structural stability. I believe that equation 1 would be more realistic if the authors computed the total energies of the reagents of a hypothetical synthesis. The possible reagents would be based on the available information of the Fe-PTC synthesis. Another possibility is to use the total energies of the products of the thermal decomposition (if they are known).

8) Regarding the adsorption energies and barriers, it would be better to present them as kJ mol-1. kJ mol-1 is unit of choice for adsorption. The authors can keep the eV unit, however including the kJ mol-1 values.

9) In case the answer to question 3 is “yes”, did the authors verify if the membrane area is significantly affected by the presence of a guest molecule? The main text should contain this information.

10) With respect to equation 2, are the total energies of the molecules (Egas) and pristine membrane (Emem) computed from the geometry-optimized structures of the isolated components or from single-point calculations of the respective geometries of the gas+membrane system?

11) The comparison between the adsorption energies for both membranes suggests that the cations (Fe and Ni) do not play an important role on the stabilization of the guest molecules on the surface. The authors need to extend the discussion about the structural features of the adsorption to include the main interactions between the guest molecules and each atom of the membrane. For instance, what are the closest distances between the guest molecules and the membrane? Are the molecules close to the cations? If yes, what is the distance between them?

12) The authors define the transition state as the following: ETS denotes the energy of a gas molecule and the Fe/Ni-PTC membrane at the transition state (namely, the energy maximum state)”. Although the highest energy value between a path corresponds to the transition state, it is by no means a maximum state. The authors must substitute the term “maximum state” for “saddle point”.

13) Even though the electron density maps clearly show that there is a stronger electrostatic repulsion between Ne/CO2/N2/Ar/CH4 and the membrane pore than the He/membrane system, the construction of an electron density difference may explain the different barriers of the former group of molecules. In the case of the electron density difference maps, three calculations should be performed: the electron density of the whole system, the guest molecule and the empty membrane. The last two quantities are subtracted from the electron density of the whole system. The maps are then built by generating isosurfaces with positive and negative isovalues (0.01 Å-3 may be adequate for your study). The positive and negative isosurfaces indicate regions where there is charge surplus and charge deficit, respectively. Dipole-shaped interactions may appear in the case of the N2 and CO2 TS structures.

14) “With the increase of temperature, the diffusion rate of gas molecules uniformly decreases, which can be attributed to the increased kinetic energy of gas molecules in the high-temperature region”. I failed to understand this sentence. Did the authors mean that, as the temperature increases, the diffusion rates increase asymptotically?

15) “For industrial He separation, the operation is normally carried out at room temperature.A citation must be added to this sentence.

16) The authors presented a very interesting computational experiment in section 3.5, resembling a separation process for industrial applications. For this reason, the results depicted in Figures 4 b and c should be presented as gas flow along the direction perpendicular to the membrane. The gas flow can be simply calculated by the time derivative of the curves of Figures 4 b and c, divided by the membrane area.
